# Racial, ethnic, and age disparities in the association of mental health symptoms and polysubstance use among persons in HIV care

Thibaut Davy-Mendez[1,2¤]*, Varada Sarovar[2], Tory Levine-Hall[2], Alexandra N. Lea[2], Amy S. Leibowitz[2], Mitchell N. Luu[3], Jason A. Flamm[4], C. Bradley Hare[5], Jaime Dumoit Smith[1], Esti Iturralde[2], James Dilley[1], Michael J. Silverberg[2], Derek D. Satre[1,2]

1 Department of Psychiatry and Behavioral Sciences, Weill Institute for Neurosciences, University of California, San Francisco, CA, United States of America, 2 Division of Research, Kaiser Permanente Northern California, Oakland, CA, United States of America, 3 Oakland Medical Center, Kaiser Permanente Northern California, Oakland, CA, United States of America, 4 Sacramento Medical Center, Kaiser Permanente Northern California, Sacramento, CA, United States of America, 5 San Francisco Medical Center, Kaiser Permanente Northern California, San Francisco, CA, United States of America

¤ Current address: Division of Infectious Diseases and Department of Epidemiology, University of North Carolina at Chapel Hill, Chapel Hill, NC, United States of America

* tdavy@med.unc.edu

**Data Availability Statement:** This study used sensitive data and access has been restricted by the Kaiser Permanente Northern California (KPNC)

## Abstract

We characterized polysubstance use burden and associations with mental health problems across demographic subgroups of PWH. In 2018–2020, as part of a primary care-based intervention study, PWH in care at three medical centers in Kaiser Permanente Northern California were screened for depression (PHQ-9$\geq$10), anxiety (GAD-2$\geq$3), and substance use (Tobacco, Alcohol, Prescription medication, and other Substance use [TAPS]$\geq$1 per substance). We used Poisson regression to estimate prevalence ratios (PRs) comparing polysubstance use prevalence (TAPS$\geq$1 for $\geq$2 substances) between PWH with positive screens for depression or anxiety vs. neither, among all PWH, and stratified by race/ethnicity and age (restricted to men), adjusting for sociodemographics, CD4, and HIV load. Screened PWH (N = 2865) included 92% men, 56% White, 19% Black, and 15% Hispanic PWH, with a median age of 55 years. Overall, polysubstance use prevalence was 26.4% (95% CI 24.9%-28.1%). PWH with depression or anxiety (n = 515) had an adjusted polysubstance use PR of 1.26 (1.09–1.46) vs. PWH with neither (n = 2350). Adjusted PRs were 1.47 (1.11–1.96), 1.07 (0.74–1.54), and 1.10 (0.85–1.41) among Black, Hispanic, and White men, respectively. Adjusted PRs did not differ by age group. Interventions should consider jointly addressing mental health and substance use problems and potential drivers, e.g. stigma or socioeconomic factors.

## Introduction

Persons with HIV (PWH) have a high prevalence of tobacco, alcohol, and other substance use, which can compromise HIV care engagement and viral suppression and contribute to adverse

Institutional Review Board (IRB). Given the sensitive nature of analytic data sets, data sharing must take place through active partnerships between external researchers and the study team. External investigators can contact either study MPI (Derek D. Satre or Michael J. Silverberg) to initiate a request for study data to support new study proposals or manuscripts. Approval of such requests and initiation of collaborations will consider several criteria including but not limited to the following: proposed project must be of high scientific merit and consistent with the overall goals and objectives of the parent study; the proposed project's investigators must plan for adequate resources to effectively complete the project. In addition, external investigators can submit a request for collaboration on the Kaiser Permanente Research Collaboration Portal at https://rcp.kaiserpermanente.org/.

**Funding:** This study was supported by the National Institute on Drug Abuse (grant number R01DA043139 to M.J.S and D.D.S.; nida.nih.gov). D.D.S was supported by a grant from the National Institute on Alcohol Abuse and Alcoholism (grant number K24AA025703; niaaa.nih.gov). T.D.-M. was supported by grants from the National Institute on Drug Abuse (grant number T32DA007250; nida.nih.gov) and the National Heart, Lung, and Blood Institute (grant number K01HL169020; nhlbi.nih.gov). The funders had no role in study design, data collection and analysis, decision to publish, or preparation of the manuscript.

**Competing interests:** The authors have declared that no competing interests exist.

health outcomes [1–10]. PWH also have a high burden of mental health problems that frequently co-occur with substance use disorders (SUDs) and can complicate care and lead to worse health outcomes, such as higher mortality [2, 11–14]. Substance use can also affect mental health and vice-versa. PWH may use substances to cope with mental health symptoms, while reducing substance use is associated with lower depressive symptoms [15, 16]. PWH who use substances are less likely to receive depression treatment when they need it [13]. Evidence from the general population also shows that substance use reduces depression treatment adherence, and that depression may increase the risk of substance use relapse [17, 18]. Identifying PWH with co-occurring mental health and substance use problems is therefore essential to address both and improve health outcomes.

Few large or recent studies have examined the use of more than one substance, or polysubstance use, among PWH in clinical care [4, 16, 19–23]. Yet polysubstance use may affect up to 20% of PWH in care, and it can lead to worse adverse substance use effects, increased sexual risk behaviors, lower antiretroviral adherence, and lower SUD treatment rates [19, 21, 24–26]. Important knowledge gaps also exist related to polysubstance use among PWH. First, studies have often excluded tobacco or alcohol from polysubstance use definitions, and few have described co-used substances, such that little is known about polysubstance use combinations including all potentially harmful and commonly used substances [4, 19–22]. Second, polysubstance vs. single substance use comorbid with mental health disorders could have a different health impact, yet the association of mental health and polysubstance use problems among PWH has rarely been investigated [27]. Third, while polysubstance use is well-studied in persons at risk for HIV and vulnerable populations such as homeless persons, the demographic profile of PWH in care with polysubstance use and with comorbid mental health problems has not been well-described [24, 28–30]. In particular, mental health and substance use disparities exist between PWH of different gender, race, ethnicity, and age [19, 31, 32]. To inform care and understand racial, ethnic, and age-based health disparities among PWH, there is a need to characterize polysubstance use, co-occurrence with mental health problems, and PWH subgroups at greater risk of these.

In this study, we leveraged electronic screenings for mental health and substance use from an HIV primary care-based intervention to characterize in detail the prevalence and patterns of polysubstance use in a clinical population of PWH, estimate the association of depression and anxiety with polysubstance use, and determine if that association varied by race, ethnicity, and age.

## Methods

### Study setting and design

We used data from Promoting Access to Care Engagement (PACE), an intervention study evaluating electronic screening and treatment for depression, anxiety, and substance use disorders in HIV primary care [33]. PACE was based at three medical centers in Oakland, Sacramento, and San Francisco in Kaiser Permanente Northern California (KPNC), a private, not-for-profit, integrated health system providing care to >4.5 million members demographically similar to other privately insured adults in the region [34, 35].

As part of PACE, from 30 October 2018 to 17 July 2020, PWH ≥18 years attending outpatient primary care visits completed self-administered questionnaires every six months, via secure messaging or in clinic on a tablet. Results were displayed in the electronic health record (EHR) to enable providers to address problems and refer patients to behavioral health specialists. PACE was approved by the Institutional Review Boards of KPNC and the University of California, San Francisco with waivers of informed consent.

We conducted a cross-sectional, secondary data analysis of PWH who completed the initial screening questionnaire during PACE.

## Study measures

Participants completed three questionnaires: the Patient Health Questionnaire (PHQ)-9, Generalized Anxiety Disorder (GAD)-2, and Tobacco, Alcohol, Prescription medication, and other Substance use (TAPS) Tool [36–38]. PACE defined a PHQ-9 score ≥10 as a positive screen for depression and GAD-2 score ≥3 as a positive screen for anxiety, based on validated thresholds [39, 40]. In this study, we divided PWH by mental health screening results: (1) PWH who screened positive for depression, anxiety, or both, and (2) PWH who screened negative for both.

The TAPS Tool captures 8 different substances: alcohol, tobacco, cannabis, illicit stimulants, heroin, prescription opioids, prescription sedatives, and prescription stimulants [38]. For each substance, TAPS includes a screener question with 5 responses ranging from "Never" to "Daily or almost daily". For alcohol, the screener question is "In the past 12 months, how often have you had 5 or more drinks (men) / 4 or more drinks (women) containing alcohol in a day?" For other substances, the screener question is "In the past 12 months, how often have you used substance X?". If the patient reports using more frequently than "Never", they respond to additional questions on past-3-month use and substance-related problems, such as trying and failing to cut down or others expressing concern. For prescription medications, TAPS only ascertains misuse (using without a prescription or more than prescribed). For cannabis, TAPS does not differentiate between medicinal and recreational use.

TAPS provides a separate risk score for each examined substance. Scores range from 0 to 4 for alcohol and from 0 to 3 for other substances. For alcohol, a score of 1 represents endorsing using 5 or more drinks (men)/4 or more drinks (women) in one day in the past year (screener question), plus using alcohol in the past 3 months. A score >1 represents further endorsing at least one alcohol use-related problem, or using 5 or more/4 or more drinks in the past 3 months. For each other substance, a score of 1 represents endorsing use in the past 3 months, and a score >1 further endorsing at least one substance use-related problem. In these analyses, a single score for opioids was created by taking the highest of the scores for heroin and misused prescription opioids. We did not examine prescription stimulants as only seven patients endorsed prescription stimulant misuse.

In primary analyses, for each substance, we defined substance use as a TAPS score ≥1 for that substance, based on instrument validation against the modified Composite International Diagnostic Interview (CIDI) for identifying problem use [38]. Polysubstance use was defined as having a TAPS score ≥1 for two or more substances. In a sensitivity analysis, we defined polysubstance use as having a TAPS score ≥2 for two or more substances. A TAPS score ≥2 has been validated to screen for CIDI-defined SUD [38].

The other variables used in this analysis were extracted from the EHR and were sex, age, HIV risk group, race, ethnicity, CD4 cell count, HIV viral load (VL), health insurance type, and neighborhood deprivation index (NDI). NDI is a census tract-level summary measure of socioeconomic indicators, shown to be associated with health outcomes [41, 42]. NDI was calculated according to published methodology and divided into quartiles based on sample distribution [41]. Age, health insurance, and NDI were measured at screening. CD4 count and VL were the closest measurement six months prior to or following the screening date.

## Statistical analysis

We estimated polysubstance use prevalence among all screened PWH and stratified by demographic and clinical characteristics listed above. We used Poisson regression models with

robust variances to estimate prevalence ratios (PRs) comparing the polysubstance use prevalence by mental health status, adjusted for all covariates [43]. To evaluate whether associations between mental health and polysubstance varied by demographic characteristics, we conducted subgroup analyses stratified by race, ethnicity, and age group, restricting to PWH who were men and Black, Hispanic, or White, due to small numbers of women and PWH in other race and ethnicity groups in our sample. Analyses were conducted in SAS v9.4 (SAS Institute Inc., Cary, NC) using a pre-specified alpha of 0.05.

## Results

### Study sample

At 8954 eligible outpatient visits, 3904 (43%) screens were completed (S1 Fig). Of 4134 unique PWH eligible for screening, 2865 (69%) completed at least one screen and were included for analyses. PWH with no completed screen were less likely to be White (45% vs. 56%) and more likely to be Hispanic (22% vs. 15%) compared with PWH who had at least one screen (S1 Table).

Of the 2865 included PWH, most (92%) were men, 56% were White, 19% Black, 15% Hispanic, and 76% men who have sex with men (MSM) (Table 1). At screening, the median age was 55 years (interquartile range [IQR]) 46–62), the median CD4 count was 644 cells/μL (IQR 472–844), and 96% had a VL<200 copies/mL. PWH characteristics varied by race and ethnicity (S2 Table). Compared with the other racial and ethnic groups, Black PWH were less likely to be men (79% vs. 89%-96%), more likely to have heterosexual risk factor (37% vs. 12%-17%), and more likely to reside in the most deprived NDI quartile (44% vs. 18%-25%).

On the initial screen (Table 1), 515 (18%) PWH had a positive screen for depression or anxiety, including 140 with a positive screen for depression only, 146 for anxiety only, and 229 for both. PWH who screened positive for depression or anxiety were younger (median 52 vs. 56 years), less likely to have a VL<200 copies/mL (90% vs. 94%), and more likely to be covered by Medicare (29% vs. 25%) or Medicaid (9% vs. 4%), compared with PWH who screened negative for both (Table 1). Racial and ethnic distribution did not differ by mental health status.

### Polysubstance use

Polysubstance use prevalence was 26.4% overall (95% confidence interval [CI] 24.9%-28.1%) and varied by sex, race, ethnicity, age, VL, and insurance type (Fig 1, all *P*<0.05). Polysubstance use prevalence was highest among PWH who were Black (32%) or Hispanic (33%), 18–29 (61%) or 30–39 (46%) years old, had a VL ≥200 copies/mL (39%), with Medicaid (41%), or in the most deprived NDI quartile (30%). PWH with polysubstance use (N = 757) used a median of 2 substances (IQR 2–3; range 2–6), most commonly alcohol/cannabis (43%), followed by alcohol/cannabis/tobacco (12%), alcohol/tobacco (11%), and cannabis/tobacco (10%) (Fig 2).

### Associations of mental health and polysubstance use

PWH who screened positive for depression or anxiety had a higher polysubstance use prevalence than those who screened negative for both, with estimates of 34.2% vs. 24.7%, respectively, and an unadjusted PR of 1.38 (95% CI 1.20–1.59) (Table 2). The adjusted PR was 1.26 (1.09–1.46).

Among PWH who were men and Black, Hispanic, or White (N = 2371), associations between mental health and polysubstance use varied by race and ethnicity (Fig 3A). Among Black men, those screening positive for depression or anxiety vs. neither had a polysubstance use PR of 1.73 (1.34–2.24) in unadjusted analyses, and 1.47 (1.11–1.96) in adjusted analyses.

**Table 1. Demographic and clinical characteristics of 2865 persons with HIV screened for depression, anxiety, and substance use disorders, Kaiser Permanente Northern California, 2018–2020.**

| Characteristic | All patients N = 2865 | Positive mental health screening [a] | |
| --- | --- | --- | --- |
| | | Depression, anxiety, or both N = 515 | Neither N = 2350 |
| Men | 2629 (92%) | 465 (90%) | 2164 (92%) |
| Race and ethnicity | | | |
| Asian or Pacific Islander | 198 (7%) | 36 (7%) | 162 (7%) |
| Black | 558 (19%) | 107 (21%) | 451 (19%) |
| Hispanic | 419 (15%) | 70 (14%) | 349 (15%) |
| White | 1602 (56%) | 284 (55%) | 1318 (56%) |
| Other/unknown | 88 (3%) | 18 (3%) | 70 (3%) |
| Age, years | 55 (46, 62) | 52 (41, 60) | 56 (47, 63) |
| HIV risk group | | | |
| MSM | 2170 (76%) | 385 (75%) | 1785 (76%) |
| IDU | 185 (6%) | 45 (9%) | 140 (6%) |
| Heterosexual or other | 510 (18%) | 85 (17%) | 425 (18%) |
| CD4 count, [b] cells/μL | 644 (472, 844) | 644 (468, 857) | 644 (474, 841) |
| HIV RNA <200 copies/mL [b] | 2678 (96%) | 466 (90%) | 2212 (94%) |
| Insurance type | | | |
| Private | 1968 (69%) | 318 (62%) | 1650 (70%) |
| Medicare | 732 (26%) | 148 (29%) | 584 (25%) |
| Medicaid | 142 (5%) | 44 (9%) | 98 (4%) |
| Other | 23 (1%) | 5 (1%) | 18 (1%) |
| NDI quartile [c] | | | |
| 1 (least deprived) | 717 (25%) | 111 (22%) | 606 (26%) |
| 2 | 726 (25%) | 143 (28%) | 583 (25%) |
| 3 | 704 (25%) | 127 (25%) | 577 (25%) |
| 4 (most deprived) | 712 (25%) | 133 (26%) | 579 (25%) |
| TAPS score ≥1 [d] | | | |
| Alcohol | 1101 (38%) | 209 (41%) | 892 (38%) |
| Cannabis | 988 (34%) | 215 (42%) | 773 (33%) |
| Tobacco | 457 (16%) | 115 (22%) | 342 (15%) |
| Stimulant | 174 (6%) | 58 (11%) | 116 (5%) |
| Sedative | 37 (1%) | 11 (2%) | 26 (1%) |
| Opioid | 23 (1%) | 7 (1%) | 16 (1%) |

Numbers are N (%) or median (IQR). Abbreviations: IDU, injection drug use; IQR, interquartile range; MSM, men who have sex with men; NDI, neighborhood deprivation index; TAPS, Tobacco, Alcohol, Prescription medication, and other Substance Use tool.

[a] Depression was defined as a PHQ-9 score ≥10. Anxiety was defined as a GAD-2 score ≥3.

[b] Closest measurement within six months before or after screening date. CD4 count was missing for 468 patients (74 [14%] with depression, anxiety, or both, and 394 [17%] without) and HIV RNA was missing for 77 patients (17 [3%] with depression, anxiety, or both, and 60 [3%] without).

[c] Calculated according to Messer et al. and divided in quartiles based on the distribution of the entire patient sample. NDI was missing for six patients [41].

[d] A TAPS score ≥1 has been validated to screen for problem use against the modified Composite International Diagnostic Interview (CIDI) [38].

The adjusted PR was 1.07 (0.74–1.54) among Hispanic men and 1.10 (0.85–1.41) among White men. For all three race/ethnicity groups, lower age was associated with higher polysubstance use prevalence, and age was the strongest confounder of the association between mental health screening and polysubstance use, with a change-in-estimate ranging 8.0%–11.4% (S3 and S4 Tables).

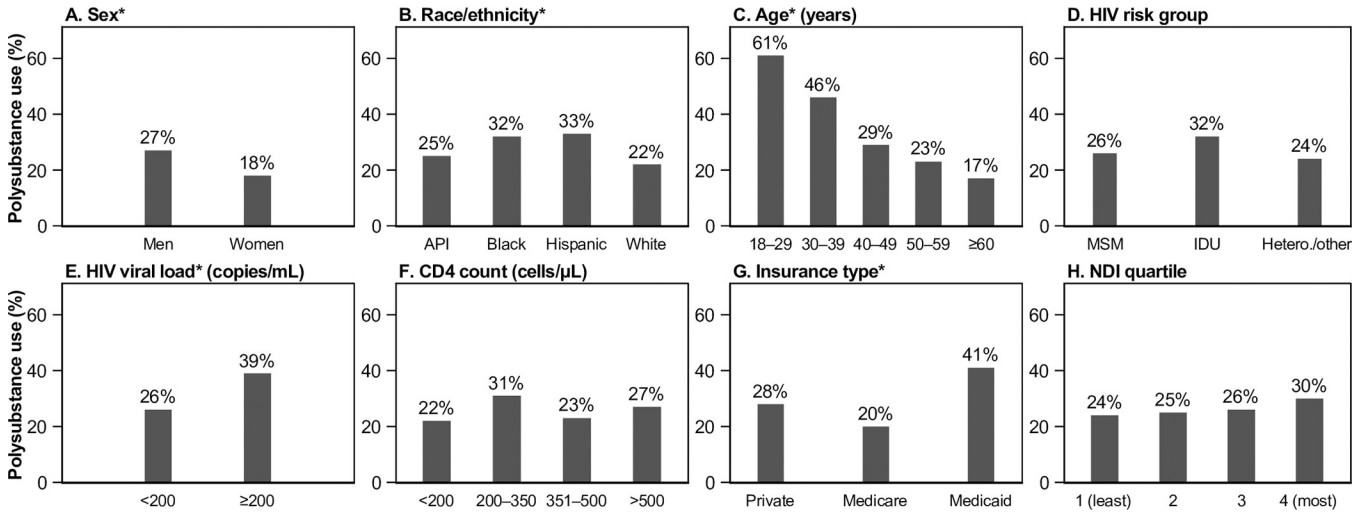

**Fig 1. Prevalence of polysubstance use stratified by clinical and demographic characteristics, among 2865 persons with HIV in Kaiser Permanente Northern California, 2018–2020.** Polysubstance use was defined as a TAPS score ≥1 for two or more substances. Abbreviations: API, Asian or Pacific Islander; Hetero., heterosexual; IDU, injection drug use; MSM, men who have sex with men. * denotes $P<0.05$ from $X^2$ tests.

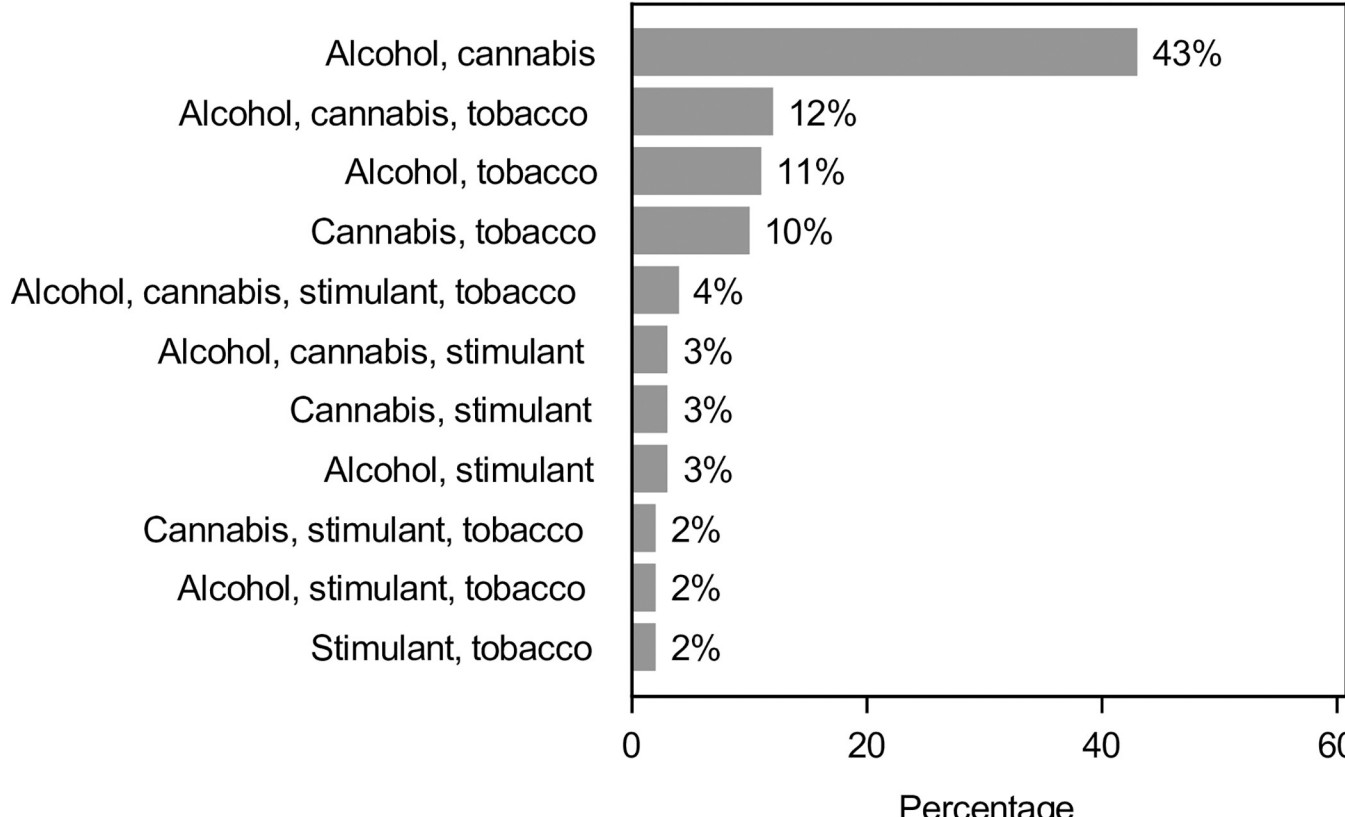

**Fig 2. Polysubstance combinations among 757 persons with HIV screening positive for problem substance use for two or more substances.** A positive screen was defined as a score ≥1 on the Tobacco, Alcohol, Prescription medication and other Substance use (TAPS) questionnaire. Only combinations accounting for >1% of patients are displayed.

**Table 2. Polysubstance prevalence and prevalence ratios comparing persons with HIV by mental health screening results, Kaiser Permanente Northern California, 2018–2020.**

| Positive mental health screen [a] | N | Polysubstance Use Prevalence [b] (95% CI) | Unadjusted PR (95% CI) [c] | Adjusted PR (95% CI) [c, d] |
|---|---|---|---|---|
| Depression, anxiety, or both | 515 | 34.2% (30.3%, 38.5%) | 1.38 (1.20, 1.59) | 1.26 (1.09, 1.46) |
| Neither | 2350 | 24.7% (23.0%, 26.5%) | 1 (ref.) | 1 (ref.) |

Abbreviations: CI, confidence interval; PR, prevalence ratio.

[a] Depression was defined as a PHQ-9 score ≥10. Anxiety was defined as a GAD-2 score ≥3.

[b] Polysubstance use was defined as having a score ≥1 for two or more substances on the Tobacco, Alcohol, Prescription medication and other Substance use (TAPS) Tool.

[c] Estimates and 95% CIs were obtained from a Poisson regression model with robust variance.

[d] Adjusted for sex, race, ethnicity, age, HIV risk group, HIV viral load, CD4 count, Neighborhood Deprivation Index (NDI), and insurance.

In the same subgroup of Black, Hispanic, or White men, among those aged ≥60 years (Fig 3B), there was no difference in polysubstance use prevalence by mental health status (adjusted PR 1.06; 95% CI 0.67–1.68). Among other age groups, we observed small differences with wide CIs, for example with an adjusted PR of 1.26 (0.91–1.75) among PWH aged 30–39 years. An adjusted PR could not be estimated for PWH aged 18–29 years.

## Sensitivity analysis

Using a TAPS score ≥2 to identify substance use, polysubstance use prevalence was 7.8% (6.8%-8.8%) overall, and this varied by race, ethnicity, age, HIV risk group, VL, insurance type, and NDI (S2 Fig, all $P<0.05$). Polysubstance use prevalence was highest for PWH who were Black (12%), 18–29 (23%) or 30–39 (15%) years old, with injection drug use risk factor (15%), VL≥200 copies/mL (17%), or Medicaid coverage (17%).

In this sensitivity analysis, screening positive for depression or anxiety was more strongly associated with polysubstance use, with an adjusted PR of 2.05 (95% CI 1.54–2.72) (S5 Table). In subgroup analyses (S3 Fig), the strongest unadjusted associations between having depression or anxiety and polysubstance use were observed among Black men (PR 3.14; 95% CI 1.99–4.94) and men aged 40–49 (PR 3.04; 95% CI 1.71–5.39). In adjusted analyses, the PR among Black men was 2.45 (95% CI 1.42–4.23), and the adjusted PR among most age groups was not estimable due to small sample sizes.

## Discussion

Among >2800 PWH screened in primary care, polysubstance use prevalence was 26%, most commonly with alcohol/cannabis, and highest among men, younger, Black, and Hispanic PWH. Having depression or anxiety was associated with a 26% increase in adjusted polysubstance use prevalence among all PWH, but a 47% increase among Black men.

Studies in PWH have used various polysubstance use definitions. In CNICS, a multisite US clinical cohort, 20% of PWH screened positive for two or more SUDs including alcohol and illicit drugs, using the Alcohol Use Disorder Identification Test-Consumption (AUDIT-C) and Alcohol, Smoking and Substance Involvement Screening Test (ASSIST) [19]. In ASTRA, a study of MSM with HIV in the United Kingdom, 51% reported any substance use, of whom 68% used two or more, or approximately a 35% polysubstance use prevalence, examining cannabis, stimulants, and opioids, amyl nitrites, and erectile dysfunction medications, but not alcohol [20]. In a Washington, DC HIV cohort, 8% were diagnosed with two or more SUDs; this study did not examine cannabis [22]. None of these studies examined tobacco use.

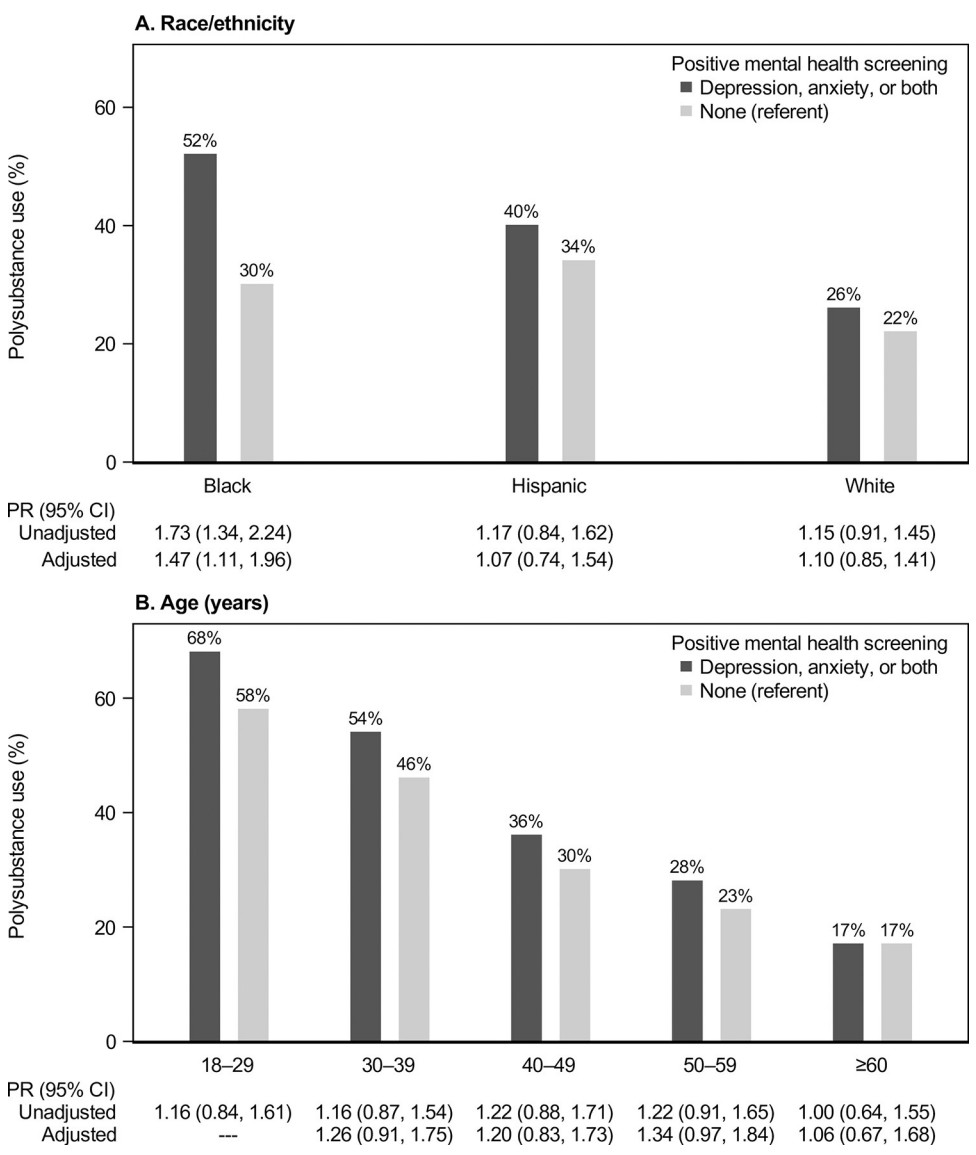

**Fig 3.** Polysubstance prevalence and prevalence ratios (PRs) with 95% confidence intervals (CIs), comparing patients by mental health screening, stratified by (A) race and ethnicity and (B) age, among 2371 men with HIV who were Black, Hispanic, or White. Adjusted models included HIV risk group, CD4 count, viral load, NDI quartile, and insurance type; models stratified by race and ethnicity are further adjusted for age, and vice versa. Numbers not displayed could not be estimated due to small sample size.

This study extends prior work by providing recent polysubstance use estimates from a large clinical sample of PWH, including all potentially harmful legal and illicit substances and detailing combinations. Our findings highlight the importance of including tobacco, alcohol, and cannabis in polysubstance use studies, as they are commonly used by PWH and can lead to considerable disease burden and adverse outcomes [1, 6, 7, 10]. In this study, the two most frequent polysubstance use combinations, together accounting for 55% of PWH with polysubstance use, included both alcohol and cannabis. Prior work has shown that the prevalence of substance use is generally highest for alcohol and cannabis, individually [3, 19]. However, past studies that have examined polysubstance use in PWH have shown a greater preponderance of stimulant use [20–22]. Several polysubstance use combinations in our study included

stimulants, together accounting for almost 20% of PWH with polysubstance use. These differences may be explained in part by geographic variations in substance use [3]. Future studies should further examine polysubstance use typologies among PWH and evaluate whether they have different mental health risk profiles.

This study also used a novel instrument, TAPS, developed as a screening tool brief enough for primary care and sufficiently detailed to inform clinical decisions [38]. We examined polysubstance use based both on TAPS score cut-offs of ≥1 and ≥2 for each substance, which identify possible problem use and possible SUD, respectively. With the higher cut-off, which may be most comparably to the CNICS study definition, polysubstance use prevalence was 8%, suggesting that higher-risk polysubstance use may be less common in our sample than in some other clinical settings [19].

We found that screening positive for depression or anxiety was associated with polysubstance use. While studies have reported associations between mental health and single or any substance use [12, 14, 44, 45], there have been few reports on mental health and polysubstance use in PWH. In one study of PWH aged 18–24 years, those with polysubstance use had greater mental health symptoms, including depression and anxiety [27]. In our study, mental health-polysubstance use associations were also stronger when using the higher cut-off of a TAPS score ≥2 for a given substance, indicating possible SUD. Clinical management of PWH with both mental health and SUDs is more complex than those with either [46]. Mental health and SUDs can have a greater impact together, including reduced HIV care engagement, lower viral suppression, and higher mortality [11, 12, 47]. Integrated screening strategies and co-management may prove highly beneficial, as addressing one issue can improve other conditions [15, 48]. Co-occurring mental health and polysubstance use problems therefore have important implications for HIV, mental health, and SUD treatment outcomes.

Our findings highlighted disparities by race and ethnicity. Polysubstance use prevalence was highest for Black and Hispanic PWH, consistent with some prior reports [22, 23]. We also found that screening positive for depression or anxiety was more strongly associated with polysubstance use among Black than among White or Hispanic men. There are several potential explanations. Black PWH are less likely to receive mental health and SUD treatment than White PWH [31, 49]. Untreated mental health problems could contribute to greater polysubstance use as a coping strategy, and untreated SUD to more severe mental health symptoms. Structural factors, like experiences of racism and socioeconomic deprivation, may also be drivers of both mental health symptoms and polysubstance use among Black men. In our study, Black PWH were more likely from deprived neighborhoods and covered by Medicaid. Black men may also experience stronger forms of stigma related to HIV status or being MSM [50, 51]. In addition to addressing mental health and substance use problems jointly, addressing drivers, for example via trauma-informed care, may provide additional benefits for Black or other PWH of color.

In our study, polysubstance use prevalence was also higher among younger PWH. CNICS and ASTRA reported that lower age was associated with substance use, including polysubstance use in ASTRA [19, 20]. In contrast, in the DC cohort, substance use was associated with older age [22]. Our findings highlight the urgency of addressing polysubstance use in younger PWH, with 61% prevalence in those aged 18–29 years when using a TAPS score of ≥1 for each substance, and 23% when using a score ≥2, which suggests a quarter of PWH in this age group may be experiencing ≥2 SUDs. Nonetheless, screening and treatment are needed at all ages among PWH. Prevalence estimates (with TAPS≥1) were 23% and 17% for those aged 50–59 and ≥60 years in our study, respectively. Older PWH may also have unique drivers of polysubstance use, such as chronic pain, social isolation, or survivor guilt. Despite the association of age with polysubstance use, we did not find strong evidence that mental health-polysubstance

use associations varied by age, indicating that PWH with depression or anxiety in a given age group in our study were not more likely to have concurrent polysubstance use than PWH with depression or anxiety in a different age group.

This study had a large and racially and ethnically diverse sample of PWH in clinical care at three KPNC sites in Oakland, Sacramento, and San Francisco, three of the 50 priority jurisdictions of the national End the HIV Epidemic (EHE) initiative [52]. Given that mental health and polysubstance use problems can compromise viral suppression and increase sexual risk behaviors [8–10, 21, 26], our findings provide important evidence that can inform EHE efforts in these and similar priority jurisdictions. Our data are recent and captured several months of COVID-19-related social distancing and shelter-in-place policies that could have exacerbated mental health and substance use problems [53]. We leveraged a rich data source including mental health and substance use screens and EHR data on sociodemographic and clinical characteristics, including NDI based on geocoded residence. Another strength is the use of the novel TAPS tool, which identifies both problem use and SUD for all potentially harmful substances including tobacco and is promoted by the NIH for implementation in health settings [54]. The implementation of EHR-integrated screening tools such as TAPS at primary care visits shows promise for increasing screening, identification, and counseling of substance use problems [55].

One important consideration is that there were few women in our study, and subgroup analyses only examined men. While our overall sample was representative of PWH in Northern California, our findings may not be generalizable to women with HIV or PWH receiving care in other settings. In addition, 31% of eligible PWH did not complete a screen. High response rates are important to provide adequate care to PWH. Future qualitative analyses from the PACE trial may shed light on drivers of non-response in this patient population. In a recent study, PWH who refused to complete a screening questionnaire cited not believing it was needed or useful as one of the reasons for non-completion; however, >80% of those who completed it felt it improved their care [56]. Communication with patients, including via providers, may help impress the importance of screening measures to improve care quality and their own satisfaction. While a completion level of 31% is similar to that of other primary care-based screening studies [55, 57], our findings could have been impacted. If PWH who did not complete a screen were more likely to have mental health and substance use problems, our findings could underestimate their prevalence in this population, particularly among Hispanic PWH, as PWH without a screen were more likely to be Hispanic. Furthermore, the estimated association between mental health and polysubstance use among Hispanic men with HIV may have been biased as a result, most likely towards the null.

Another important consideration is that our study used a cross-sectional design to characterize associations of mental health and polysubstance use problems. Longitudinal studies are needed to better understand the link between mental health and polysubstance use in PWH, including the degree to which mental health symptoms may lead to polysubstance use, e.g. as a coping strategy, polysubstance use may worsen mental health symptoms, and common drivers may contribute to both worse mental health symptoms and increased polysubstance use.

## Conclusions

PWH in KPNC had high polysubstance use prevalence which was associated with having elevated depression or anxiety symptoms, especially among Black men. Research and interventions should jointly consider mental health and substance use problems as well as possible drivers of both conditions.

## Supporting information

**S1 Table. Characteristics of 4134 persons with HIV eligible for screening, stratified by screening completion, Kaiser Permanente Northern California, 2018–2020.**
(DOCX)

**S2 Table. Demographic and clinical characteristics stratified by race and ethnicity.**
(DOCX)

**S3 Table. Unadjusted and adjusted prevalence ratios (95% confidence intervals) for the association with polysubstance use, stratified by race/ethnicity, among men with HIV.**
(DOCX)

**S4 Table. Prevalence ratios comparing the probability of polysubstance use between PWH with positive screens for depression, anxiety, or both and PWH with no positive mental health screen, unadjusted and adjusted for a single covariate at a time, with the percentage change from unadjusted estimate.**
(DOCX)

**S5 Table. Polysubstance use prevalence and prevalence ratios comparing persons with HIV by mental health screening results, in a sensitivity analysis using a TAPS score $\geq 2$.**
(DOCX)

**S1 Fig. Flowchart of patient inclusion.**
(DOCX)

**S2 Fig. Prevalence of polysubstance use by clinical and demographic characteristics in a sensitivity analysis using a TAPS score $\geq 2$.** Abbreviations: API, Asian or Pacific Islander; Hetero., heterosexual; IDU, injection drug use; MSM, men who have sex with men; NDI, neighborhood deprivation index. * denotes $P < 0.05$.
(DOCX)

**S3 Fig.** Prevalence of polysubstance among patients screening positive for either depression or anxiety versus neither, stratified by (A) race and ethnicity and (B) age, in a sensitivity analysis using a TAPS score $\geq 2$.
(DOCX)

## Author Contributions

**Conceptualization:** Thibaut Davy-Mendez, Varada Sarovar, Tory Levine-Hall, Alexandra N. Lea, Amy S. Leibowitz, Mitchell N. Luu, Jason A. Flamm, C. Bradley Hare, Jaime Dumoit Smith, Esti Iturralde, James Dilley, Michael J. Silverberg, Derek D. Satre.

**Data curation:** Thibaut Davy-Mendez, Varada Sarovar, Tory Levine-Hall.

**Formal analysis:** Thibaut Davy-Mendez, Varada Sarovar, Tory Levine-Hall.

**Funding acquisition:** James Dilley, Michael J. Silverberg, Derek D. Satre.

**Investigation:** Amy S. Leibowitz, Mitchell N. Luu, Jason A. Flamm, C. Bradley Hare, Esti Iturralde, James Dilley, Michael J. Silverberg, Derek D. Satre.

**Project administration:** Alexandra N. Lea, Amy S. Leibowitz, Jaime Dumoit Smith, Michael J. Silverberg, Derek D. Satre.

**Supervision:** Alexandra N. Lea, Mitchell N. Luu, Jason A. Flamm, C. Bradley Hare, Jaime Dumoit Smith, Michael J. Silverberg, Derek D. Satre.

**Writing – original draft:** Thibaut Davy-Mendez, James Dilley, Derek D. Satre.

**Writing – review & editing:** Thibaut Davy-Mendez, Varada Sarovar, Tory Levine-Hall, Alexandra N. Lea, Amy S. Leibowitz, Mitchell N. Luu, Jason A. Flamm, C. Bradley Hare, Jaime Dumoit Smith, Esti Iturralde, James Dilley, Michael J. Silverberg, Derek D. Satre.

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
