## [Decision Letter · Decision Letter 0]

4 Sep 2023

PONE-D-23-17520Racial, ethnic, and age disparities in the association of mental health symptoms and polysubstance use among persons in HIV carePLOS ONE

Dear Dr. Davy-Mendez,

Thank you for submitting your manuscript to PLOS ONE. After careful consideration, we feel that it has merit but does not fully meet PLOS ONE’s publication criteria as it currently stands. Therefore, we invite you to submit a revised version of the manuscript that addresses the points raised during the review process. I regret that we were not able to secure a second reviewer for this manuscript. Thus, in the interest of time (the manuscript was submitted many months ago), I, as Academic Editor, undertook the second review.  My expertise in substance use research, mental health, and my independence from any contributing authors qualify me for this.   This is a nice manuscript that describes baseline characteristics of a cohort being followed as part of an intervention to implement, assess efficacy, and cost-effectiveness of electronic screening modalities for mental health in HIV primary care clinics at three Kaiser Permanente Northern California locations.  I agree with Reviewer 1 that the manuscript provides noteworthy and actionable information, especially among persons engaged in care.  This largely descriptive manuscript adds to the literature regarding the prevalence of substances used and indicators of potential excess risk of polysubstance use.  One recognizes that the 'actionable' factors will be further explored as the trial progresses and ends.  To that end, I disagree with several of Reviewer #1's suggestions, including conducting latent class analysis, bidirectional models, and causal inference methods.  The papers' contribution is to describe the population and many of the factors this paper is exploring will either be done as part of outcome analyses of the trial.  Latent class analysis is valuable to identify patient populations that have similar symptom profiles and have the potential ot further inform the field.  However, I believe that this analysis may be more valuable when examining symptom profiles that contribute to the low (or high) efficacy of the intervention. Further, the measures used in this study to screen for mental health symptoms and substance use are being increasingly used in primary health settings across the country. LCA may help to identify homogeneous groups, but are not likely to be useful for providers in busy primary care settings who likely don't have the time to categorize patients in these groups, whereas the screening instruments have defined cut-offs, and validated performance characteristics. LCA may be interesting to conduce when the trial is over and additional exploratory analyses are conducted with the trial outcomes. In my mind this paper does accomplish the main goal of describing the prevalence and excess risk of polysubstance use in the sample. For these reasons, I also do not endorse assessing bidirectional models, or using inverse probability weighting at this stage. This may be reflective of my own training in epidemiology. I applaud the authors for presenting prevalence and prevalence ratios and value this as a contribution to the field.  Some minor comments: 1. I also agree that the findings regarding Black patients are important.  I agree that some additional analyses would be relevant here, however my suggestion is to assess what is confounding the aPRs in the assocation between having depression/anxiety and polysubstance use: what factors made the adjusted risk estimates so much higher?  2. I also agree that some additional brief information regarding the TAPs instrument would be helpful.  Especially as the authors believe that this is a novel factors of the study.3. Please clarify if there were measures of opioid use in the survey of substances. 4. It is not clear why the authors refer to the the data being collected in the priority jurisdictions of the End the HIV Epidemic initiative. How does this study related to that initiative? Thank you for your patience in the processing of this manuscript. 

Please submit your revised manuscript by Oct 19 2023 11:59PM. If you will need more time than this to complete your revisions, please reply to this message or contact the journal office at plosone@plos.org. Please include the following items when submitting your revised manuscript:A rebuttal letter that responds to each point raised by the academic editor and reviewer(s). You should upload this letter as a separate file labeled 'Response to Reviewers'.A marked-up copy of your manuscript that highlights changes made to the original version. You should upload this as a separate file labeled 'Revised Manuscript with Track Changes'.An unmarked version of your revised paper without tracked changes. You should upload this as a separate file labeled 'Manuscript'.If applicable, we recommend that you deposit your laboratory protocols in protocols.io to enhance the reproducibility of your results. Protocols.io assigns your protocol its own identifier (DOI) so that it can be cited independently in the future. For instructions see: https://journals.plos.org/plosone/s/submission-guidelines#loc-laboratory-protocols. Additionally, PLOS ONE offers an option for publishing peer-reviewed Lab Protocol articles, which describe protocols hosted on protocols.io. Read more information on sharing protocols at https://plos.org/protocols?utm_medium=editorial-email&utm_source=authorletters&utm_campaign=protocols.

We look forward to receiving your revised manuscript.

Kind regards,

Kimberly Page, PhD, MPH

Academic Editor

PLOS ONE

“This study was supported by the National Institute on Drug Abuse (grant numbers R01DA043139, T32DA007250). D.D.S was supported by a grant from the National Institute on Alcohol Abuse and Alcoholism (K24AA025703).”

 “No”

“No”

Reviewers' comments:

Reviewer's Responses to Questions

**Comments to the Author**

1. Is the manuscript technically sound, and do the data support the conclusions?

Reviewer #1: Yes

2. Has the statistical analysis been performed appropriately and rigorously? 

Reviewer #1: Yes

3. Have the authors made all data underlying the findings in their manuscript fully available?

Reviewer #1: Yes

4. Is the manuscript presented in an intelligible fashion and written in standard English?

Reviewer #1: Yes

5. Review Comments to the Author

Reviewer #1: This is an interesting manuscript describing the association of screening positive for a mental health disorder (i.e., depression or anxiety) with polysubstance use assessed via the TAPS. There are several strengths from this large sample of PWH who are engaged in care that provide opportunities to guide expanded efforts to optimize screening and treatment for co-occurring mental health and substance use disorders. Below I provide some feedback that may assist with further refining this contribution.

1) TAPS - It would be useful to have a bit more information regarding the TAPS measure and some indication for what the thresholds of 1 (Problem Use) and 2 (Higher Risk) are indexing for each substance. I was also confused by the >= 1 at various points in the manuscript, but it became clear that this was for two substances. Consider referring to the TAPS total scores (i.e., >= 2 and >=4) at some point to make this more clear.

2) Analysis - The analytic approach is generally sound but one wonders if a Latent Class Analysis would not provide more nuanced information regarding polysubstance use typologies that can then be examined as correlates. Figure 2 was particularly helpful in this regard but LCA would assist with empirically derived typologies.

3) Stimulants - It seems worth noting that 15% of participants had some polysubstance pattern that involved stimulants and this is without assessing substances that are commonly co-used among MSM (e.g., GHB, amyl nitrites, ketamine, erectile dysfunction medications). Stimulants are strongly associated with depression and other forms of negative affect, and again one wonders if there are distinct polysubstance use typologies that are driving observed associations with mental health indicators.

4) Negative Reinforcement Models - The authors note that substance may be used as a means of coping, consistent with the self-medication hypothesis. It may also be that using substances induces symptoms of withdrawal (including depression and anxiety). Addressing the potentially bi-directional connections between substance use and mental health seems worthwhile.

5) Black Race - In my opinion, these are the most interesting and timely findings. It would be interesting to know how the social determinants of health examined in this study (e.g., Area Deprivation Index, Medicaid) could account for or moderate observed associations with Black race. It would be useful to have a table reported the bivariate and adjusted associations for each of the models (i.e., completing screening and polysubstance use outcomes).

6) Screening Rates - It is noteworthy that approximately one-third of potentially eligible individuals did not complete the screening measures, and some systemic predictors of completing screening were identified. This seems to be an important finding in and of itself for two reasons. First, from a clinical perspective we need patients to complete these measures to even identify that they are in need of treatment for mental health or substance use disorders. Some discussion of ways to optimize screening efforts such as contingency management (patients) of pay for performance (providers) would be useful. Second, the issue of non-response introduces systematic biases in efforts to estimate the prevalence of polysubstance use and its association with mental health. Consider, deploying causal inference methods to adjust for predictors of non-response (e.g., inverse probability weighting).

6. PLOS authors have the option to publish the peer review history of their article (what does this mean?). If published, this will include your full peer review and any attached files.

Reviewer #1: **Yes: **Adam Carrico

---

## [Author Response · Author response to Decision Letter 0]

20 Oct 2023

EDITOR COMMENTS

This is a nice manuscript that describes baseline characteristics of a cohort being followed as part of an intervention to implement, assess efficacy, and cost-effectiveness of electronic screening modalities for mental health in HIV primary care clinics at three Kaiser Permanente Northern California locations. I agree with Reviewer 1 that the manuscript provides noteworthy and actionable information, especially among persons engaged in care. This largely descriptive manuscript adds to the literature regarding the prevalence of substances used and indicators of potential excess risk of polysubstance use. One recognizes that the 'actionable' factors will be further explored as the trial progresses and ends. 

To that end, I disagree with several of Reviewer #1's suggestions, including conducting latent class analysis, bidirectional models, and causal inference methods. The papers' contribution is to describe the population and many of the factors this paper is exploring will either be done as part of outcome analyses of the trial. Latent class analysis is valuable to identify patient populations that have similar symptom profiles and have the potential ot further inform the field. However, I believe that this analysis may be more valuable when examining symptom profiles that contribute to the low (or high) efficacy of the intervention. Further, the measures used in this study to screen for mental health symptoms and substance use are being increasingly used in primary health settings across the country. LCA may help to identify homogeneous groups, but are not likely to be useful for providers in busy primary care settings who likely don't have the time to categorize patients in these groups, whereas the screening instruments have defined cut-offs, and validated performance characteristics. LCA may be interesting to conduce when the trial is over and additional exploratory analyses are conducted with the trial outcomes. In my mind this paper does accomplish the main goal of describing the prevalence and excess risk of polysubstance use in the sample. For these reasons, I also do not endorse assessing bidirectional models, or using inverse probability weighting at this stage. This may be reflective of my own training in epidemiology. I applaud the authors for presenting prevalence and prevalence ratios and value this as a contribution to the field. 

Thank you for the feedback regarding the contribution of the analyses conducted in this manuscript and actionable implications for patients engaged in care. We are also grateful for the helpful suggestions for improvement, and the guidance regarding analyses suggested by the Reviewer 1. The analyses suggested by the Reviewer 1 such as LCA could yield informative findings and we plan to integrate such approaches in future analyses using outcome data from the completed trial. Please see also our specific replies to Reviewer 1, below. 

Some minor comments:

1. I also agree that the findings regarding Black patients are important. I agree that some additional analyses would be relevant here, however my suggestion is to assess what is confounding the aPRs in the assocation between having depression/anxiety and polysubstance use: what factors made the adjusted risk estimates so much higher? 

We agree that the findings by race/ethnicity merit additional detail in the manuscript given their importance. We have added two supplemental tables. S3 Table displays unadjusted and adjusted prevalence ratios (PRs) for all covariates, stratified by race/ethnicity. S4 Table displays the change in PR estimate for the association of mental health and polysubstance use, adjusting for only one covariate at a time, stratified by race/ethnicity. In addition, we have added the following statements to describe these findings:

Results, page 11, lines 243–246:

“For all three race/ethnicity groups, lower age was associated with higher polysubstance use prevalence, and age was the strongest confounder of the association between mental health screening and polysubstance use, with a change-in-estimate ranging 8.0%–11.4% (S3 and S4 Tables).”

2. I also agree that some additional brief information regarding the TAPs instrument would be helpful. Especially as the authors believe that this is a novel factors of the study.

We have added additional details on the TAPS instrument, including its structure and scoring system, in the Methods, pages 5–6, lines 114–142:

“The TAPS Tool captures 8 different substances: alcohol, tobacco, cannabis, illicit stimulants, heroin, prescription opioids, prescription sedatives, and prescription stimulants [38]. For each substance, TAPS includes a screener question with 5 responses ranging from “Never” to “Daily or almost daily”. For alcohol, the screener question is “In the past 12 months, how often have you had 5 or more drinks (men) / 4 or more drinks (women) containing alcohol in a day?” For other substances, the screener question is “In the past 12 months, how often have you used substance X?”. If the patient reports using more frequently than “Never”, they respond to additional questions on past-3-month use and substance-related problems, such as trying and failing to cut down or others expressing concern. For prescription medications, TAPS only ascertains misuse (using without a prescription or more than prescribed). For cannabis, TAPS does not differentiate between medicinal and recreational use.

TAPS provides a separate risk score for each examined substance. Scores range from 0 to 4 for alcohol and from 0 to 3 for other substances. For alcohol, a score of 1 represents endorsing using 5 or more drinks (men)/4 or more drinks (women) in one day in the past year (screener question), plus using alcohol in the past 3 months. A score >1 represents further endorsing at least one alcohol use-related problem, or using 5 or more/4 or more drinks in the past 3 months. For each other substance, a score of 1 represents endorsing use in the past 3 months, and a score >1 further endorsing at least one substance use-related problem. In these analyses, a single score for opioids was created by taking the highest of the scores for heroin and misused prescription opioids. We did not examine prescription stimulants as only seven patients endorsed prescription stimulant misuse.

In primary analyses, for each substance, we defined substance use as a TAPS score ≥1 for that substance, based on instrument validation against the modified Composite International Diagnostic Interview (CIDI) for identifying problem use [38]. Polysubstance use was defined as having a TAPS score ≥1 for two or more substances. In a sensitivity analysis, we defined polysubstance use as having a TAPS score ≥2 for two or more substances. A TAPS score ≥2 has been validated to screen for CIDI-defined SUD [38].”

3. Please clarify if there were measures of opioid use in the survey of substances. 

We have added details about how TAPS and our analyses capture opioid use in the revisions described in response to Editor comment #2. These are highlighted below:

Methods, page 5, line 114:

“The TAPS Tool captures 8 different substances: alcohol, tobacco, cannabis, illicit stimulants, heroin, prescription opioids, prescription sedatives, and prescription stimulants.38”

Methods, page 6, line 122:

“For prescription medications, TAPS only ascertains misuse (using without a prescription or more than prescribed).”

Methods, page 6, line 132:

“In these analyses, a single score for opioids was created by taking the highest of the scores for heroin and misused prescription opioids.”

4. It is not clear why the authors refer to the data being collected in the priority jurisdictions of the End the HIV Epidemic initiative. How does this study related to that initiative? 

We have clarified this point in the Discussion, pages 15–16, lines 359–364:

“This study had a large and racially and ethnically diverse sample of PWH in clinical care at three KPNC sites in Oakland, Sacramento, and San Francisco, three of the 50 priority jurisdictions of the national End the HIV Epidemic (EHE) initiative [52]. Given that mental health and polysubstance use problems can compromise viral suppression and increase sexual risk behaviors [8-10, 21, 26], our findings provide important evidence that can inform EHE efforts in these and similar priority jurisdictions.”

REVIEWER COMMENTS

Reviewer #1: This is an interesting manuscript describing the association of screening positive for a mental health disorder (i.e., depression or anxiety) with polysubstance use assessed via the TAPS. There are several strengths from this large sample of PWH who are engaged in care that provide opportunities to guide expanded efforts to optimize screening and treatment for co-occurring mental health and substance use disorders. Below I provide some feedback that may assist with further refining this contribution.

1) TAPS - It would be useful to have a bit more information regarding the TAPS measure and some indication for what the thresholds of 1 (Problem Use) and 2 (Higher Risk) are indexing for each substance. I was also confused by the >= 1 at various points in the manuscript, but it became clear that this was for two substances. Consider referring to the TAPS total scores (i.e., >= 2 and >=4) at some point to make this more clear.

We thank the reviewer for their comments and suggestions. We have clarified the language around TAPS scoring throughout the manuscript. We have also added additional details on the TAPS instrument in the Methods. See our response to the Editor’s comment #2 above.

2) Analysis - The analytic approach is generally sound but one wonders if a Latent Class Analysis would not provide more nuanced information regarding polysubstance use typologies that can then be examined as correlates. Figure 2 was particularly helpful in this regard but LCA would assist with empirically derived typologies.

The primary objective of this study was to characterize associations of mental health and polysubstance use problems across demographic groups at baseline in the PACE trial. Details on the combinations of substances used by participants are provided in Figure 2 for descriptive purposes. We agree with the reviewer that more nuanced typologies of polysubstance use would be informative, and we appreciate the suggestion of suing LCA to examine these typologies. Following guidance from the Editor, and given that examining these typologies fell outside the scope of the present study, we plan to incorporate LCA approaches and examinations of typologies in future analyses that will include follow up screening and outcome data from the trial. See also our response to the Editor, above. We have added statements regarding the importance of polysubstance use typologies in response to this comment and comment #3 (see below).

3) Stimulants - It seems worth noting that 15% of participants had some polysubstance pattern that involved stimulants and this is without assessing substances that are commonly co-used among MSM (e.g., GHB, amyl nitrites, ketamine, erectile dysfunction medications). Stimulants are strongly associated with depression and other forms of negative affect, and again one wonders if there are distinct polysubstance use typologies that are driving observed associations with mental health indicators.

We have expanded the following paragraph in the Discussion section, page 13, lines 292–305:

“This study extends prior work by providing recent polysubstance use estimates from a large clinical sample of PWH, including all potentially harmful legal and illicit substances and detailing combinations. Our findings highlight the importance of including tobacco, alcohol, and cannabis in polysubstance use studies, as they are commonly used by PWH and can lead to considerable disease burden and adverse outcomes [1, 6, 7, 10]. In this study, the two most frequent polysubstance use combinations, together accounting for 55% of PWH with polysubstance use, included both alcohol and cannabis. Prior work has shown that the prevalence of substance use is generally highest for alcohol and cannabis, individually [3, 19]. However, past studies that have examined polysubstance use in PWH have shown a greater preponderance of stimulant use [20-22]. Several polysubstance use combinations in our study included stimulants, together accounting for almost 20% of PWH with polysubstance use. These differences may be explained in part by geographic variations in substance use [3]. Future studies should further examine polysubstance use typologies among PWH and evaluate whether they have different mental health risk profiles.”

4) Negative Reinforcement Models - The authors note that substance may be used as a means of coping, consistent with the self-medication hypothesis. It may also be that using substances induces symptoms of withdrawal (including depression and anxiety). Addressing the potentially bi-directional connections between substance use and mental health seems worthwhile.

We agree that the potential bi-directional connections between substance use and mental are important to highlight. Given the cross-sectional nature of this study, we could not examine the timing of mental health relative to substance use and vice versa. We have added the following statements to the Discussion, page 17, lines 393–398:

“Another important consideration is that our study used a cross-sectional design to characterize associations of mental health and polysubstance use problems. Longitudinal studies are needed to better understand the link between mental health and polysubstance use in PWH, including the degree to which mental health symptoms may lead to polysubstance use, e.g. as a coping strategy, polysubstance use may worsen mental health symptoms, and common drivers may contribute to both worse mental health symptoms and increased polysubstance use.”

5) Black Race - In my opinion, these are the most interesting and timely findings. It would be interesting to know how the social determinants of health examined in this study (e.g., Area Deprivation Index, Medicaid) could account for or moderate observed associations with Black race. It would be useful to have a table reported the bivariate and adjusted associations for each of the models (i.e., completing screening and polysubstance use outcomes).

We have added two supplemental tables. S3 Table displays unadjusted and adjusted prevalence ratios (PRs) for all covariates, stratified by race/ethnicity. S4 Table displays the change in PR estimate for the association of mental health and polysubstance use, adjusting for only one covariate at a time, stratified by race/ethnicity. As part of the planned future analyses of this hybrid implementation/effectiveness trial, we will examine how computerized screening (versus usual care by providers) may have resulted in increased identification of substance use and mental health problems. We will also examine whether patient factors, including but not limited to neighborhood deprivation, age, and race/ethnicity, were associated with receipt of screening and behavioral interventions, and these analyses are ongoing. For this reason, examining associations with completing a screen was not an objective of the current analysis. Differences between patients who were screened vs. not screened were provided descriptively in S1 Table. No additional tables were added as there were no relevant modeling analyses to report.

We have added the following statements to describe the findings reported in S3 and S4:

Results, page 11, lines 243–246:

“For all three race/ethnicity groups, lower age was associated with higher polysubstance use prevalence, and age was the strongest confounder of the association between mental health screening and polysubstance use, with a change-in-estimate ranging 8.0%–11.4% (S3 and S4 Tables).”

6) Screening Rates - It is noteworthy that approximately one-third of potentially eligible individuals did not complete the screening measures, and some systemic predictors of completing screening were identified. This seems to be an important finding in and of itself for two reasons. First, from a clinical perspective we need patients to complete these measures to even identify that they are in need of treatment for mental health or substance use disorders. Some discussion of ways to optimize screening efforts such as contingency management (patients) of pay for performance (providers) would be useful. Second, the issue of non-response introduces systematic biases in efforts to estimate the prevalence of polysubstance use and its association with mental health. Consider, deploying causal inference methods to adjust for predictors of non-response (e.g., inverse probability weighting).

We agree with the reviewer that understanding patterns of screening are important, and the PACE trial includes analyses of qualitative interviews as well as quantitative data on screening completion. As noted in response #5, these analyses are in process. Given the cross-sectional design and largely descriptive nature of our study objective, we feel that methods intended to answer causal questions using longitudinal data are outside the scope of this study. However, we agree with the reviewer about the important implications of non-response both for clinical care and potential limitations of our analyses. We have expanded the section on non-response, including discussion of possible bias, in the Discussion, pages 16–17, lines 378–391:

“In addition, 31% of eligible PWH did not complete a screen. High response rates are important to provide adequate care to PWH. Future qualitative analyses from the PACE trial may shed light on drivers of non-response in this patient population. In a recent study, PWH who refused to complete a screening questionnaire cited not believing it was needed or useful as one of the reasons for non-completion; however, >80% of those who completed it felt it improved their care [56]. Communication with patients, including via providers, may help impress the importance of screening measures to improve care quality and their own satisfaction. While a completion level of 31% is similar to that of other primary care-based screening studies [55, 57], our findings could have been impacted. If PWH who did not complete a screen were more likely to have mental health and substance use problems, our findings could underestimate their prevalence in this population, particularly among Hispanic PWH, as PWH without a screen were more likely to be Hispanic. Furthermore, the estimated association between mental health and polysubstance use among Hispanic men with HIV may have been biased as a result, most likely towards the null.”

---

## [Editor Report · Decision Letter 1]

2 Nov 2023

Racial, ethnic, and age disparities in the association of mental health symptoms and polysubstance use among persons in HIV care

PONE-D-23-17520R1

Dear Dr. Davy-Mendez,

We’re pleased to inform you that your manuscript has been judged scientifically suitable for publication and will be formally accepted for publication once it meets all outstanding technical requirements.

Kind regards,

Kimberly Page, PhD, MPH

Academic Editor

PLOS ONE

Additional Editor Comments (optional): I commend the authors on their revisions.  I think this paper will be highly cited as the field moves forward in examining polysubstance use and the associated health outcomes. 
---

## [Editor Report · Acceptance letter]

17 Nov 2023

PONE-D-23-17520R1 

Racial, ethnic, and age disparities in the association of mental health symptoms and polysubstance use among persons in HIV care 

Dear Dr. Davy-Mendez:

I'm pleased to inform you that your manuscript has been deemed suitable for publication in PLOS ONE. Congratulations! Your manuscript is now with our production department. 

Kind regards, 

on behalf of

Dr. Kimberly Page 

Academic Editor

PLOS ONE